



# Improved Counting Statistics of an Ultrafine DMPS System

Dominik Stolzenburg[1,2*], Tiia Laurila[1,*], Pasi Aalto[1], Joonas Vanhanen[3], Tuukka Petäjä[1], Juha Kangasluoma[1]

[1]Institute for Atmospheric and Earth System Research / Physics, Faculty of Science, University of Helsinki, P.O. Box 64, 00014 Helsinki, Finland
[2]Institute for Materials Chemistry, Faculty of Technical Chemistry, Technical University of Vienna, 1060 Vienna, Austria
[3]Airmodus Ltd., 00560 Helsinki, Finland
*These authors contributed equally to this work

*Correspondence to*: Juha Kangasluoma (juha.kangasluoma@helsinki.fi)

**Abstract.** Differential mobility particle size spectrometers (DMPS) are widely used to measure the aerosol number size-distribution. Especially during new particle formation (NPF) the dynamics of the ultrafine size-distribution determine the significance of the newly formed particles within the atmospheric system. A precision quantification of the size-distribution and derived quantities such as new particle formation and growth rates is therefore essential. However, size-distribution measurements in the sub-10 nm range suffer from high particle losses and are often derived from only a few counts in the DMPS system, making them subject to very high counting uncertainties. Here we show that a CPC (modified Airmodus A20) with a significantly higher aerosol optics flow rate compared to conventional ultrafine CPCs can greatly enhance the counting statistics in that size-range. Using Monte Carlo uncertainty estimates, we show that the uncertainties of the derived formation and growth rates can be reduced from 10-20% down to 1% by deployment of the high statistics CPC on a strong NPF event day. For weaker events and hence lower number concentrations, the counting statistics can result in a complete breakdown of the growth rate estimate with relative uncertainties as high as 75%, while the improved DMPS still provides reasonable results at 10% relative accuracy. In addition, we show that other sources of uncertainty are present in CPC measurements, which might become more important when the uncertainty from the counting statistics is less dominant. Altogether, our study shows that the analysis of NPF events could be greatly improved by the availability of higher counting statistics in the used aerosol detector of DMPS systems.

## 1 Introduction

Differential/Scanning mobility particle size spectrometers (DMPS or SMPS) systems can be used to measure the number size distribution of ambient aerosol particles ranging in size from sub-10 nm to hundreds of nanometers (Aalto et al., 2001; Wang and Flagan, 1990). The instruments typically consist of an impactor, a charger, a DMA (Differential Mobility Analyzer), and a CPC (Condensation Particle Counter). The impactor is used to limit the maximum particle size to enable multiple charging corrections in the inversion. The charger then brings the particles to a known charge distribution (typically steady-state bipolar charging equilibrium as described by e.g. Wiedensohler, 1988), and the charged particles are size-selected in a DMA based on





their electrical mobility. Finally, the number concentration is counted by condensational growth and subsequent optical detection with a CPC. The number size-distribution is then determined by stepping/scanning different voltages at the DMA and the application of an inversion process if the maximum particle size, the charging probability, all the losses, and the detection efficiency are known.

As size predominantly determines the dynamics of ultrafine aerosol particles, measurements of the particle number size distribution are essential for understanding the role of aerosols in the atmospheric system. One process were the smallest ultrafine (<100 nm) particles are most important is so-called atmospheric new particle formation (NPF). During NPF, small molecular clusters form from gaseous precursors and subsequently grow to larger sizes (Kulmala et al., 2013), where they can

contribute to the budget of cloud condensation nuclei and impact the Earth's radiative balance (e.g. Gordon et al., 2017). To obtain an in-depth understanding of the dynamics of NPF, it is essential to measure the number size-distribution down to even sub-10 nm aerosol particles accurately and reliably (Dada et al., 2020; Kulmala et al., 2012). There is still room for technological developments in the instrumentation aimed for the detection of the sub-10 nm particle size distributions, which help to reduce discrepancies between data sets (Kangasluoma et al., 2020). In the sub-10 nm size range, a large fraction of the

sample is lost in the measurement system due to diffusional losses, low charging probability, and low detection efficiency of the CPC especially in the sub-5 nm size range, emphasizing the need to acquire sufficient statistics for the counted particles.

The number of registered counts in the CPC are determined from the total size-dependent penetration of the DMPS/SMPS, the CPC aerosol flow rate through the optics and the sampling interval for an individual size. This can be summarized within a PI parameter to characterize an SMPS system in the sub-10 nm size range (Cai et al., 2019). The main idea behind the PI parameter

is that when the sampling time is constant, an instrument with larger PI parameter will obtain larger number of counted particles for the same inlet aerosol concentration and hence it describes the instrument sensitivity towards low number concentrations. Most recent advances in the sub-10 nm size distribution instrumentation have been focused on increasing the sampling time (Stolzenburg et al. 2017), size resolution (Kangasluoma et al. 2018), or inversion performance (Stolzenburg et al., 2022). However, in terms of the PI parameter, large advances are expected simply by using a CPC with a large aerosol flow rate,

which linearly increases the number of counted particles. In addition, it remains unquantified to what extend improved counting statistics provide more reliable results on quantities typically inferred from sub-10 nm size-distributions, such as the particle growth and formation rate. Solid uncertainty estimates for these size distribution-derived quantities are rare (Dada et al., 2020; Kangasluoma and Kontkanen, 2017) or only provided via sophisticated inversion schemes (Ozon et al., 2021).

In the current work, we use a new laminar flow CPC (modified Airmodus A20) that has 2.5 lpm aerosol (and optics) flow rate

within a DMPS (Kangasluoma et al., 2015). It was operated in Hyytiälä, Finland, in parallel with a TSI 3776 as the detector downstream of the same DMPS system. Here, we demonstrate the improved data quality given by the larger counting statistics, perform an uncertainty analysis for the system, and finally determine the effect of the counting statistics on the calculations of the particle growth and formation rate through Monte Carlo analysis.


## 2 Methods

### 2.1 Measurement setup

The measurements were performed from 24.03.2017-19.05.2017 at the SMEAR II station (Station for Measuring Ecosystem-Atmosphere Relations; Hari & Kulmala, 2005). The station is located in Southern Finland, Hyytiälä (61$^o$51'N, 24$^o$17'E). The DMPS system used in this measurement has a short Hauke-type DMA that was used to select particle sizes in the range of 1-40 (Aalto et al., 2001) and operated at an aerosol to sheath flow ratio of 4 lpm/ 20 lpm. The modified Airmodus A20 CPC and the TSI 3776 CPC measured in parallel in the DMPS system as illustrated in Fig. 1. In parallel of this nano-DMPS setup, a long-DMPS using a different DMA (long-column Hauke DMA), but the same inlet and charger was operated simultaneously (Aalto et al., 2001). In addition, also the total aerosol number concentration above 4 nm is determined using a TSI 3775 CPC sampling outside air without an upstream DMA.

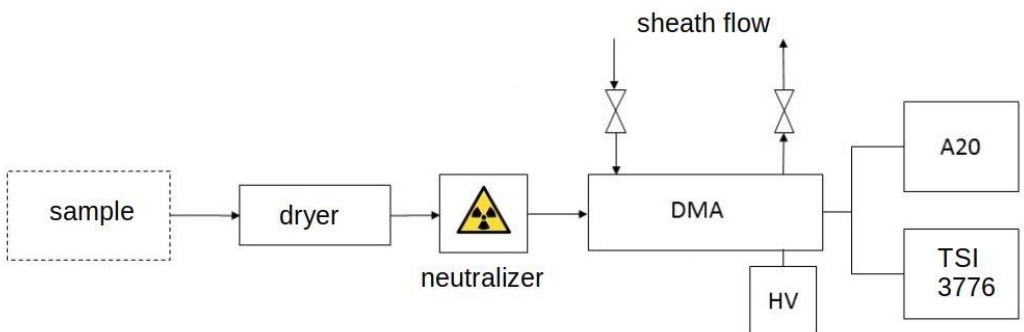

**Figure 1.** Schematic of the measurement setup. Sample is taken from the ambient air and dried to < 40% relative humidity, neutralized with a bipolar diffusion charger, and a DMA is used for size classification, operated at an aerosol to sheath flow ratio of 4 lpm/20 lpm. Downstream of the DMA the sample is split between the two CPCs.

### 2.2 CPCs

The modified Airmodus A20 CPC is a laminar flow CPC, where the entire sample flow is heated and saturated with butanol. The saturated sample flow goes to a multi-tube (6 tubes) condenser, where the temperature is decreased to activate the aerosol particle growth by condensation, followed by optical detection. The nominal cut-off diameter, using the factory settings, of the Airmodus A20 CPC is 7 nm. The TSI 3776 CPC is also a laminar-type CPC, but in contrast to the A20 CPC, the TSI 3776 CPC utilizes the ultrafine CPC design, where the sample flow is introduced in the middle of the condenser with a capillary (M. R. Stolzenburg & McMurry, 1991). The TSI 3776 CPC was operated with the high-flow setting, where the CPC draws an inlet



flow of 1.5 lpm, of which 1.2 lpm is directed to a bypass. Of the remaining 0.3 lpm, 0.25 lpm is used as a sheath flow and 0.05 lpm as the sample flow, i.e. the effective detector flow of undiluted sample in the CPC optics are only 0.05 lpm.

Kangasluoma et al. (2015) showed that a conventional, unsheathed CPC can be tuned for even sub-3 nm particle detection by increasing the temperature difference between the saturator and the condenser, and by adjusting the inlet flow rate. With the

factory settings of the Airmodus A20 CPC, the saturator temperature is 39 °C and condenser temperature is 15 °C. The modified Airmodus A20 CPC used in this study has saturator temperature of 44 °C and condenser temperature of 10 °C, and the inlet flow rate was increased from 1 lpm to 2.5 lpm, which is entirely analysed in the optics unit, resulting in a factor of 50 difference in analyzed sample flow between the two CPCs.

The detection efficiency of the CPCs was characterized using negative silver particles produced with a tube furnace. The test

particles were charged with a [241]Am radioactive source and size classified with a short Hauke DMA running at aerosol flow rate of 4 lpm and sheath flow rate of 20 lpm. The CPCs, the TSI 3776, modified A20 and standard A20, were calibrated one by one against a TSI electrometer 3068B running at 1 lpm flow rate.

Figure S1 in the Supplement shows the cut-off calibration curves for the CPCs. The 50% cut-off diameters of the Airmodus A20, the modified Airmodus A20 and the TSI 3776 CPC are approximately 5.5 nm, 2.9 nm and 2.0 nm, respectively. With the

modifications the modified Airmodus A20 CPC has a performance almost comparable to the TSI 3776 CPC. It should be noted, that this specific device in this specific calibration performed exceptionally well, as its nominal cutoff is typically closer to 2.5-3 nm for silver test particles (Wlasits et al., 2020).

## 2.3 Counting process of a CPC: Poisson process

The particles are counted in the optical unit of the CPC, where a nozzle directs the particle stream to cross a laser beam

perpendicularly. Light is scattered from the laser beam as the particles cross it, and the scattered light is collected by a photodiode. In typical optics with ~1 lpm aerosol flow, the probability of coincidence in the counting process is negligible with moderate number concentrations ($< 30\,000$ $cm^{-3}$), which are typically measured downstream of a DMPS system. The measured particle number concentration ($cm^{-3}$) is determined by the number of particles counted in a specific measurement volume and the number concentration $C$ can be calculated by using the volumetric flow rate through the optics $Q_{opt}$, the

measurement time $t$ and the counted particles $N$:

$$C = \frac{N}{Q_{opt} \cdot \tau} \quad (1)$$

In our setup, we can neglect the total penetration of the system since the compared CPCs measure in parallel in the same DMPS system and the total penetration is the same for both. This allows us to compare the raw data from the CPCs without an inversion and the uncertainties related to it (Stolzenburg et al., 2022). Rearranging Eq. (1) for the counted particles $N$, shows

that we can predict that a factor 50 increase of $Q_{opt}$ (effective undiluted optics flow of 0.05 lpm in the TSI 3776 versus 2.5 lpm in the modified Airmodus A20) should lead to a factor 50 increase of N:

$$N = C \cdot \tau \cdot Q_{opt} \quad (2)$$



A random variable N has a Poisson distribution with the parameter $\mu\tau > 0$, where $\tau$ is the measurement time, and $\mu$ is the intensity of the process, if the random variable can obtain discrete values (0,1,2,3,...) within the time interval $\tau$ with a

probability $P(N,\tau)$ as in Eq. (3)

$$P(N,\tau) = \frac{e^{-\mu\tau}(\mu\tau)^N}{N!}$$
(3)

A process can be considered a Poisson process if it fulfils the following requirements.

1. In separate finite time intervals $\tau$, the numbers of events detected obey the Poisson distribution $P(N,\tau)$
2. In separate time intervals, the numbers of detected events are independent of each other

3. The probability of coincidence is negligible

According to the above requirements, a CPC counting process can be considered a Poisson process. A Poisson distribution can be shown to have the following properties: the expected value $< N >$ of the distribution can be calculated as $< N >= \mu\tau$, and the standard deviation ($\sigma$) can be calculated as $\sigma = \sqrt{\mu\tau} = \sqrt{< N >}$. Thus, the uncertainty resulting from the counting statistics of a CPC can be calculated as the square root of the expected value of the particle count.

**2.4 Uncertainty in CPC measurements**

Uncertainty is a fundamental concept in statistics and probability, and it occurs in all measurements. The uncertainty of a measurement can be systematic, due to human error or random resulting from the natural fluctuation of the observed system. In most cases, the total uncertainty of the measurement is a combination of uncertainty from multiple sources. Ultimately, we are interested in the uncertainty of the data obtained from an individual CPC within a DMPS setup, which

could be used within uncertainty estimates of subsequently derived variables (J, GR). However, we are typically not able to quantify the total uncertainty and are not able disentangle the counting process from other sources of uncertainty, such as electronic noise or flow variations in the CPC optics (called measurement error in the following). However, our specific setup allows us to confine the counting uncertainty due to the availability of another CPC.

We chose the following approach to obtain an uncertainty estimate of the measurements with the DMPS using the TSI 3776

as a detector. First, only data for particles $\geq$ 6 nm are used for the error analysis to ensure that the detection efficiencies of the CPCs do not affect the result. As we can see in Fig. S1, at 6 nm, the calibration curves of both CPCs have plateaued. Next, we choose the measurement time where the modified Airmodus A20 measures particle counts $N$ in certain narrow ranges [$N_1$, $N_2$], where $N_2=1.05 \cdot N_1$ with $N_1 \leq N \leq N_2$. The counts from the corresponding times are then selected from the parallel measuring TSI 3776. These selected particle counts are plotted as a normalized histogram, and a Gaussian probability density

function (PDF) is fitted to the data. Figure 2 shows four examples of the resulting histograms and fits.





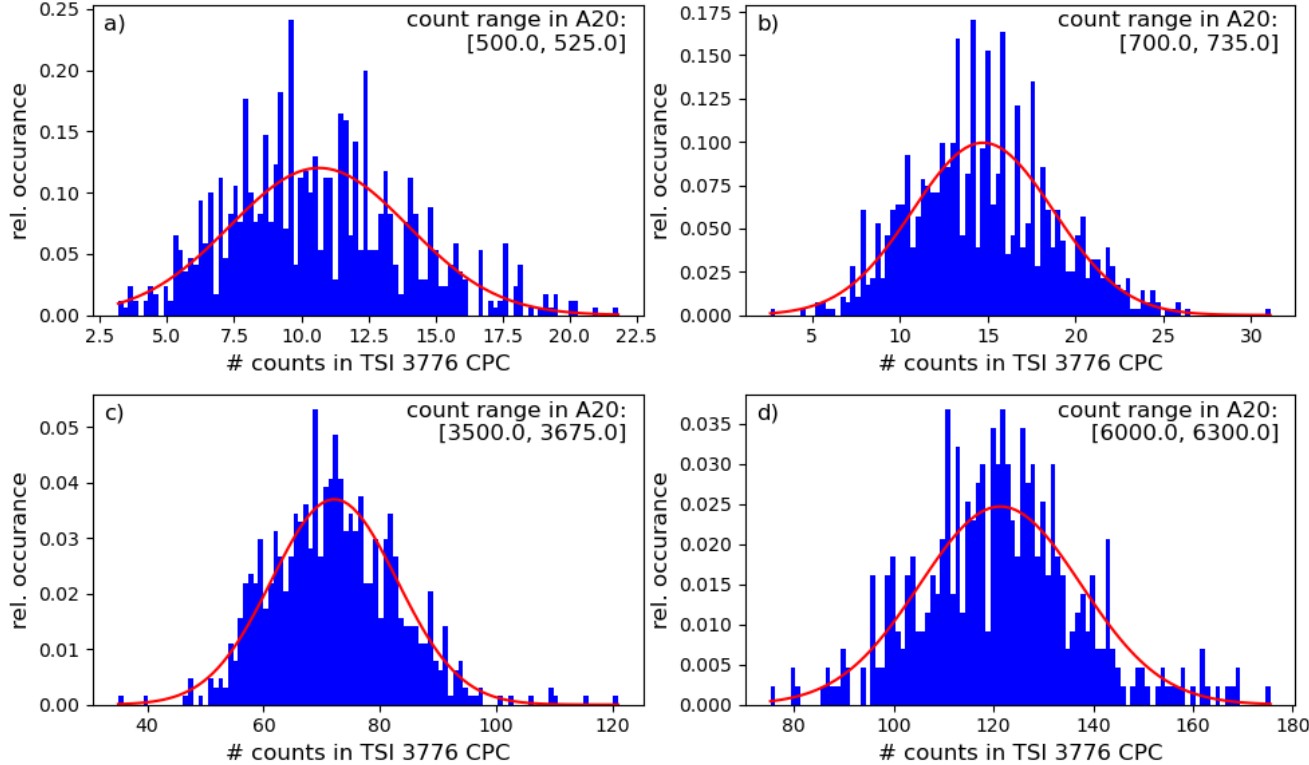

**Figure 2** Distributions of TSI 3776 counts from four example count ranges simultaneously measured in the modified A20 (a) [500, 525], (b) [700, 735], (c) [3500, 3675], (d) [6000, 6300]. The counts from the same times are selected from the TSI 3776 CPC and plotted as a histogram. Red line shows the fitted Gaussian PDF.

We are now interested in the uncertainties determining the width of these PDFs. By selecting count ranges in the modified Airmodus A20, we select measurements with an actual number concentration $C_{true} \pm \Delta C_{true}$, where the uncertainty originates from the counting and measurement error in the modified Airmodus A20 and the finite width of selected counts in the interval range, which are assumed to be independent error sources and hence can be expressed in relative uncertainties as follows:

$$\frac{\Delta C_{true}}{C_{true}} = \sqrt{\left(\frac{\Delta N_{A20}^{count}}{N_{A20}}\right)^2 + \left(\frac{\Delta N_{A20}^{meas}}{N_{A20}}\right)^2 + \left(\frac{\Delta N_{A20}^{width}}{N_{A20}}\right)^2} \qquad (4)$$

The $C_{true}$ constrained by the selection of modified Airmodus A20 measurements, is also measured simultaneously by the TSI 3776. Therefore, the PDF of counts measured in the TSI 3776 (or its width, i.e. its relative uncertainty $\Delta N_{TSI}^{PDF}/N_{TSI}$) results from the uncertainty in the $C_{true}$-values selected by the modified Airmodus A20 measurements, the counting error of the 3776 and the measurement error of the 3776, expressed in relative uncertainties as follows:

$$\frac{\Delta N_{TSI}^{PDF}}{N_{TSI}} = \sqrt{\left(\frac{\Delta N_{TSI}^{count}}{N_{TSI}}\right)^2 + \left(\frac{\Delta N_{TSI}^{meas}}{N_{TSI}}\right)^2 + \left(\frac{\Delta C_{true}}{C_{true}}\right)^2} = \sqrt{\left(\frac{\Delta N_{TSI}^{count}}{N_{TSI}}\right)^2 + \left(\frac{\Delta N_{TSI}^{meas}}{N_{TSI}}\right)^2 + \left(\frac{\Delta N_{A20}^{count}}{N_{A20}}\right)^2 + \left(\frac{\Delta N_{A20}^{meas}}{N_{A20}}\right)^2 + \left(\frac{\Delta N_{A20}^{width}}{N_{A20}}\right)^2} \quad (5)$$





We see that besides the measurement errors, we can specify all terms in Eq. (5). As we aim to determine the total error of the TSI 3776 of an independent measurement of a concentration $C_{true}$ given by the uncertainties in counting and measurement, which we can now link to the measured width of the PDF via Eq.(5) obtaining:

$$\frac{\Delta N_{TSI}^{tot}}{N_{TSI}} = \sqrt{\left(\frac{\Delta N_{TSI}^{count}}{N_{TSI}}\right)^2 + \left(\frac{\Delta N_{TSI}^{meas}}{N_{TSI}}\right)^2} = \sqrt{\left(\frac{\Delta N_{TSI}^{PDF}}{N_{TSI}}\right)^2 - \left(\frac{\Delta N_{A20}^{count}}{N_{A20}}\right)^2 - \left(\frac{\Delta N_{A20}^{meas}}{N_{A20}}\right)^2 - \left(\frac{\Delta N_{A20}^{width}}{N_{A20}}\right)^2} \qquad (6)$$

If we now neglect the uncertainty in the measurement of the modified Airmodus A20 CPC, Eq. (6) provides an upper estimate
of the total error in the CPC 3776.

**2.5 Growth rate and formation rate and propagated uncertainties via MC-simulations**

Using these error estimates, we can derive the corresponding uncertainties in the quantities typically derived from DMPS size-distribution data, the growth rate (GR) and formation rate (J). Here, we calculate the GR using the 50% appearance time method (Stolzenburg et al., 2018; Lehtipalo et al., 2014) with an automated algorithm, which after manually defining a time-
window for the NPF event, fits sigmoidal functions to the rise of the measured signal in each size channel separately. The 50% appearance times are then plotted against the sizes of the corresponding channels and a linear interpolation is used for the size range 3-6 nm (2.99-6.28 nm) and 6-10 nm (6.28-10.94 nm) to obtain $GR_{3-6}$ and $GR_{6-10}$ as the slope of that interpolation, respectively.

The formation rate can be calculated for particle size range [$d_p$, $d_p+\Delta d_p$] according to Eq. (7) (Kulmala et al. 2012):

$$J_{dp} = \frac{dN_{dp}}{dt} + CoagS_{dp}N_{dp} + \frac{GR}{\Delta dp}N_{dp} \qquad (7)$$

Here, CoagS is the coagulation sink (loss rate of particles in that size range with the background particles due to coagulation) and $N_{dp}$ is the number concentration of the particles in the size range [$d_p$, $d_p+\Delta d_p$]. The coagulation sink is calculated for geometric mean diameter of the selected size range and in the atmospheric conditions typical for the SMEAR II, it can be empirically estimated from the condensation sink of a non-volatile vapor (Dal Maso et al., 2005) as in Eq. (8) (Lehtinen et al.,
180 2007)

$$CoagS_{dp} = CS\left(\frac{dp}{0.71}\right)^{-1.6} \qquad (8)$$

We use the size-interval [3 nm, 6 nm] to calculate the formation rate at 3 nm ($J_3$) in all subsequent calculations. For the automated algorithm, the integrated concentration $N_{dp}$ of the interval was smoothed and the $GR_{3-6}$ value for the specific NPF day was used as input for the last term. The diurnal variation of $J_3$ was then fitted by a Gaussian expression and its peak value
was used as the NPF-event specific $J_3$ value.

We performed a Monte Carlo simulation on one of the NPF days (28th March 2017). New sets of data were generated from the original data 10 000 times, by altering the measured counts in each size-channel for each measurement time according to their underlying uncertainties. We performed three sets of MC simulations. First and second, we use a Poisson counting error to





vary the TSI 3776 and the modified Airmodus A20 data. The generated input data (counts) were used to directly calculate
$GR_{3-6}$ and $GR_{6-10}$ as the appearance time method can be performed on the raw signal and is independent from any inversion
procedure (Lehtipalo et al., 2014). For the calculation of the formation rate, we inverted the raw signal into a size-distribution
using a least-square algorithm which also using the data above 10 nm obtained from the long-DMPS. Comparison of the
resulting formation and growth rates allows the investigation of the effect of increasing counting statistics with respect to these
size-distribution derived quantities. As a third simulation, we assume the total error for the TSI 3776 derived via Eq. (6) (upper
error estimate) as the input uncertainty in the Monte Carlo runs altering the raw counts and compare it with the Poisson only
case of the TSI 3776 to investigate the magnitudes of counting and measurement error on $GR_{3-6}$, $GR_{6-10}$ and $J_3$.

## 3 Results

### 3.1 Effect of counting statistics on the inverted size-distributions and number closure

We analysed the dataset by classification of the NPF event days (Dal Maso et al., 2005) and calculated formation and growth
rates for the subset of class-I NPF event days. Figure 3 shows an example NPF day (28th March 2017) from both CPCs
(modified Airmodus A20 Fig. 3a and TSI 3776 Fig. 3b). We can see that the modified Airmodus A20 produces a smoother
distribution in the areas of low concentrations (blue-to-yellow color range). Besides the lower nominal cut-off in the laboratory
calibration of the TSI 3776 (Fig. S1), the signal at the small sizes below 5 nm is noisier in the TSI 3776 derived size-distribution
compared to the modified Airmodus A20 derived size-distribution. Potentially, the overall reduced statistics counterbalance
the effect of a more efficient detection at these sizes. Moreover, it needs to be noted that ambient cut-offs are subject to larger
uncertainties due to the unknown chemical composition of the counted particles and the composition-dependent response of
the CPCs (Wlasits et al., 2020).

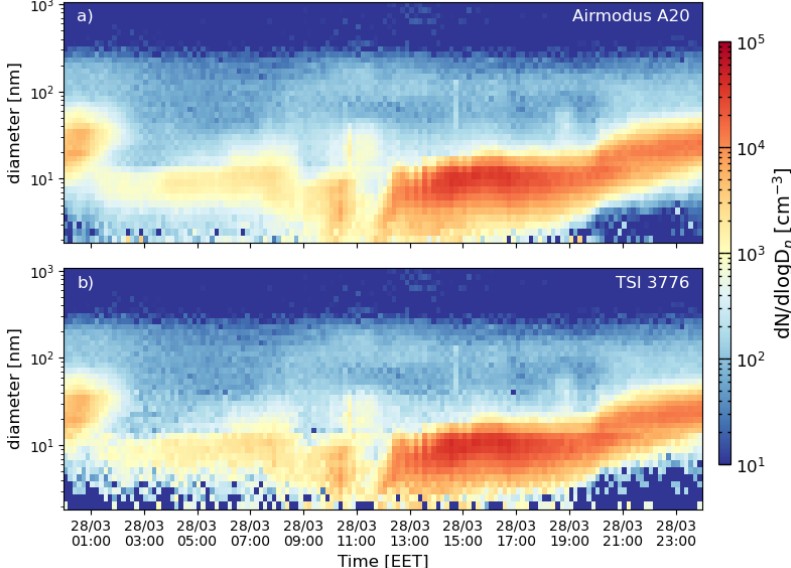

**Figure 3** Comparison of the inverted size-distribution using the signal of two different CPCs in the nano-DMPS (2-40 nm) for the 28[th] March
2017, a strong NPF day in Hyytiälä, Finland. (a) shows the size-distribution in dN/dlogDp as color-code with the measured diameter on the
y-axis and the time on the x-axis using the modified Airmodus A20 as detector in the nano-DMPS. (b) shows the same using the TSI 3776
as detector in the nano-DMPS.

Next, we compare the performance of the DMPS using different detectors with respect to the number closure with a

simultaneously measuring total CPC (TSI Model 3775, nominal cutoff 4 nm). The correlation of the full campaign dataset

between the integrated number concentration of the DMPS system (above 4 nm) and the total concentration measurement with

the CPC 3775 are shown in Fig. 4 for both detectors (Fig. 4a using the TSI 3776 in the inversion and subsequent integration

and Fig. 4b using the modified Airmodus A20). The Pearson's coefficient of correlation is high for both (0.992 and 0.994),

but slightly better in case when the modified Airmodus A20 is used within the DMPS inversion, which is reasonable due to

the increased statistics. However, the data deviates from the 1:1 relation (0.89 slope for the modified Airmodus A20, which is

more significant than for the TSI 3776 based DMPS data with a slope of 0.94). This could be due to different plateau value

reached in the counting efficiency curves or could be caused by the fact that at sizes around 4 nm the modified Airmodus A20

is more affected by uncertainties in the CPC cut-off curve (Fig. 2). Ambient cut-offs are likely to be at larger sizes due to

different chemical composition than the test particles used in the calibrations, and plateau values of the same instrument often

vary slightly between different calibrations even for the same instrument (Wlasits et al., 2020). Therefore, both reasons could

easily lead to offsets in the inversion resulting in the observed discrepancies in the total number concentration of around 10%

at high concentrations.





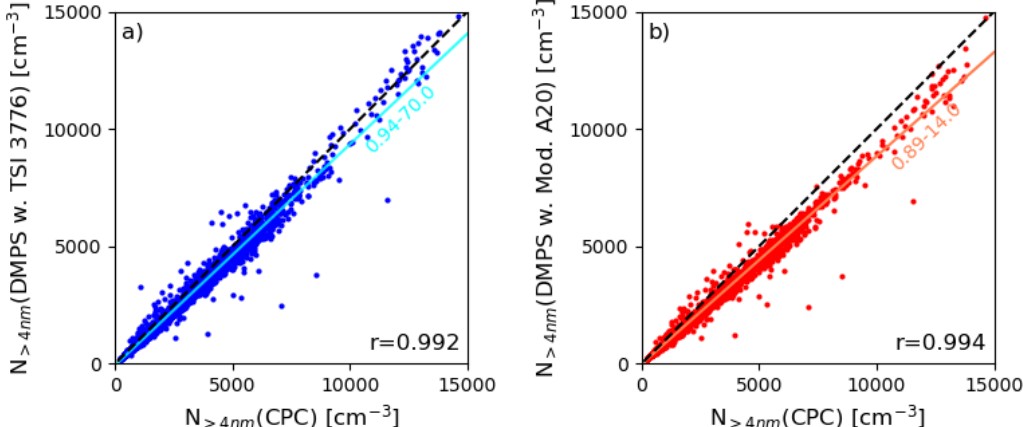

**Figure 4** Comparison of the total number concentration above 4 nm obtained from integration of the inverted DMPS data and the total concentration measurement using a TSI 3775 CPC. (a) shows the correlation for the entire campaign dataset when the TSI 3776 is used in the DMPS inversion and total concentration integration and (b) shows the same when the modified Airmodus A20 is used. The cyan and coral solid lines show the linear fit to the data indicating the deviation from the 1:1 dashed black line.

### 5.2 The effect of increased counting statistics on the particle formation and growth rates

In Fig. 5, we compare the calculated $GR_{3-6}$, $GR_{6-10}$ and $J_3$ values obtained from the DMPS data with the different underlying detectors for all NPF class-I events (see Dal Maso et al., 2005) recorded throughout the campaign (in total 19 events). We observe strong correlations in the derived growth and formation rates, with the lowest correlation coefficient for $GR_{3-6}$, where the signal is most noisy. Interestingly, the formation rate is more robust, even if derived at 3 nm, where also the $GR_{3-6}$ is used within the calculation of Eq. (7).

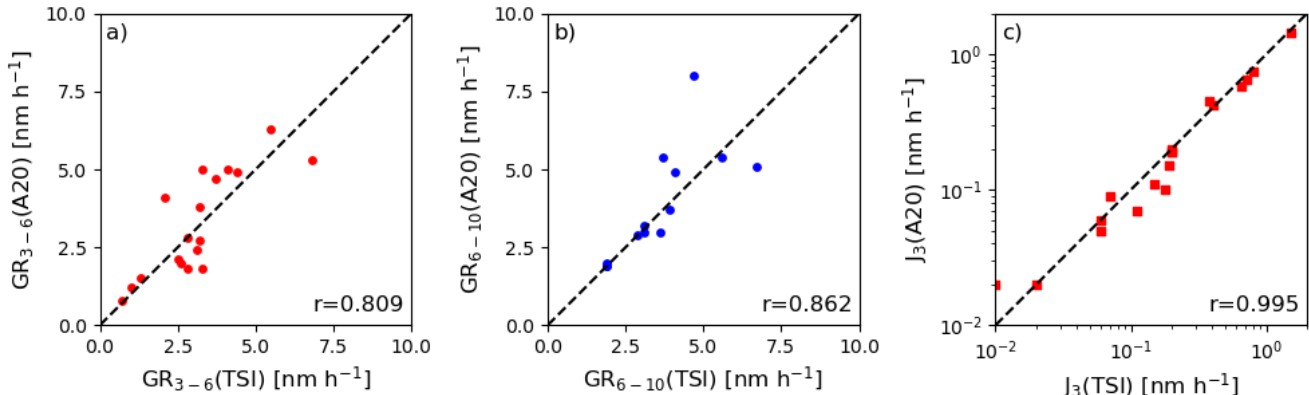

**Figure 5** Comparison of the $GR_{3-6}$ (a), $GR_{6-10}$ (b) and $J_3$ (c) obtained from the datasets recorded by the TSI 3776 (x-axes) and modified Airmodus A20 (y-axes) used as detector downstream of the same DMA.





In Fig. 6 we present the results from our Monte-Carlo analysis of the 28th March 2017 comparing the performance of the modified Airmodus A20 with the TSI3776 assuming the measured signal is only subject to a counting uncertainty. Fig 7a and 7b present the results of the 10000 $GR_{3-6}$ and $GR_{6-10}$ calculations performed with the same automated appearance time algorithm, showing the obtained 50% appearance times at each diameter (channel) on top of the original size-distribution and

the corresponding linear fits for the GR estimate. Apparently, the smaller the channel size the larger the spread between the appearance time results, especially for the TSI 3776, where the relative uncertainty of each measurement becomes very large below 4 nm due to the limited count rates (which is in the range of 10 counts per measurement during NPF). It needs to be noted that it seems to be especially the channel at 3 nm, which has a broad spread in 50% appearance times dominating the variation in the subsequent $GR_{3-6}$ derivation.

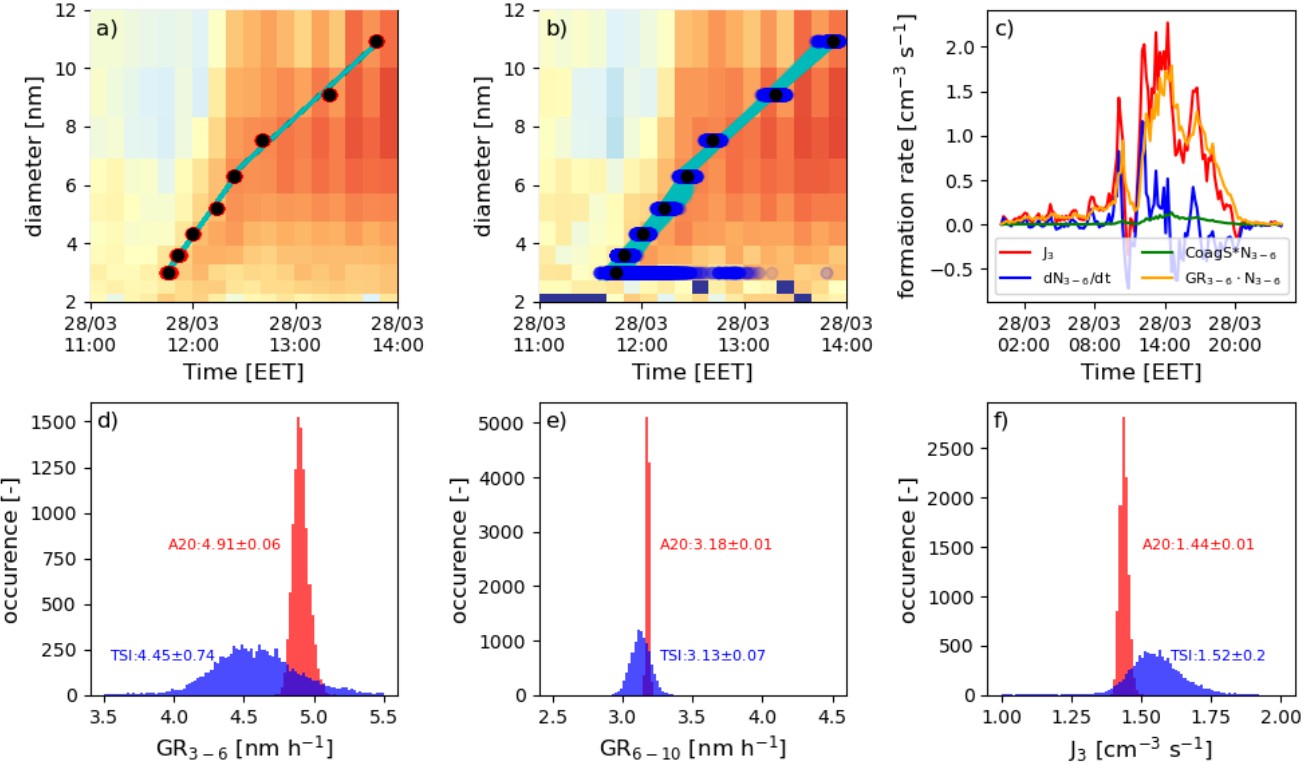


**Figure 6** Results from the Monte Carlo simulations testing the influence of a pure counting error on the size distribution-derived quantities $J_3$, $GR_{3-6}$ and $GR_{6-10}$. (a) shows the appearance times (red dots) and linear growth rate fits (for two size-ranges, cyan lines) for 10000 Monte Carlo runs on top of the original size-distribution (color code not shown and only for illustrative purposes) randomly varying the count rates in the modified Airmodus A20 (black dots result from the original data). (b) shows the same for the TSI 3776 dataset (blue dots are
appearance times derived from Monte Carlo varied data, black dots are original data and cyan lines the linear growth rate fits). (c) shows the formation rate calculation at 3 nm according to Eq. (6) with the red line $J_3$, the blue line the approximated change in total number concen of the calculation bin, the green line the correction term due to the coagulation sink, the orange line the correction term due to the growth flux out of the size-bin of interest. (d)-(f) show the histograms of the Monte Carlo results for the $GR_{3-6}$ (d), $GR_{6-10}$ (e), and $J_3$ (f), with the red histograms corresponding to values derived from the modified Airmodus A20 dataset and the blue histograms corresponding to values
derived from the TSI 3776 dataset.





This directly translates into the significantly larger variance of the $GR_{3-6}$ values derived from the TSI 3776 compared to the modified Airmodus A20 (Fig. 6d and 6e). For $GR_{3-6}$ the statistical uncertainty from the counting error is larger than 16% for the TSI 3776 compared to only 1% when using the modified Airmodus A20. $GR_{6-10}$ shows lower overall uncertainties and

less, but still significant, differences between the two CPCs (2% compared to 0.3%). Interestingly, the mean of the Monte-Carlo distributions is slightly offset between the two CPCs for both $GR_{3-6}$ and $GR_{6-10}$ demonstrating the observed variations shown in Fig. 6. However, even though we saw very good correlation for the $J_3$ values within the campaign derived from both instruments, it seems that $J_3$ is also heavily influenced by the counting statistics. In Hyytiälä, the most dominant term in the calculation of the formation rate is often the growth term out of the bin of interest, i.e. $\frac{GR}{\Delta dp}N_{dp}$ (Eq. (7) and Fig. 6c) and hence

the fluctuations in $GR_{3-6}$ are directly translated (Fig. 6f) into large uncertainties for the TSI 3776 derived $J_3$ (13% relative uncertainty) and much lower in the modified Airmodus A20 derived $J_3$ (1%).

In addition, it needs to be noted, that the 28th March 2017 was one of the days with the highest formation rate ($J_3\sim1.5$ cm$^{-3}$ s$^{-1}$) throughout the campaign. Therefore, we repeated the analysis for two additional days with significantly lower $J_3$ (5th May 2017 and 6th May 2017, with $J_3=0.05$ cm$^{-3}$ s$^{-1}$ and $J_3=0.15$ cm$^{-3}$ s$^{-1}$, respectively). We present the Monte Carlo results for $GR_{3-6}$ and

$J_3$ for the intermediate formation rate day (6th May 2017) in Fig. S2 in the Supplement. As expected, the lower $J_3$ also resulted in lower count rates in both CPCs during NPF. Therefore, also a larger counting uncertainty in the size distribution-derived quantities was observed, with up to 25% relative uncertainty in $GR_{3-6}$ and $J_3$ when the TSI 3776 is used and with a still significant reduction for the modified Airmodus A20 down to 7-8% relative uncertainty (for the 6th May 2017). At very low $J_3$ (5th May 2017, Fig. 7), the Monte Carlo distributions for the TSI 3776 data get skewed and the Monte Carlo results shows a

bimodal distribution with unphysical GR values around 0 indicating problems with the automated GR fitting. The relative uncertainty becomes as large as 75%. This shows that GR values derived at such low number concentrations and which such low counting statistics are not reliable. Only instrumentation which provides enough signal can be used: even though the modified Airmodus A20 relative uncertainty already becomes as large as 10%, this value is still lower than the relative uncertainty of $GR_{3-6}$ for the TSI dataset of a very strong NPF event day with almost two orders of magnitude higher $J_3$.




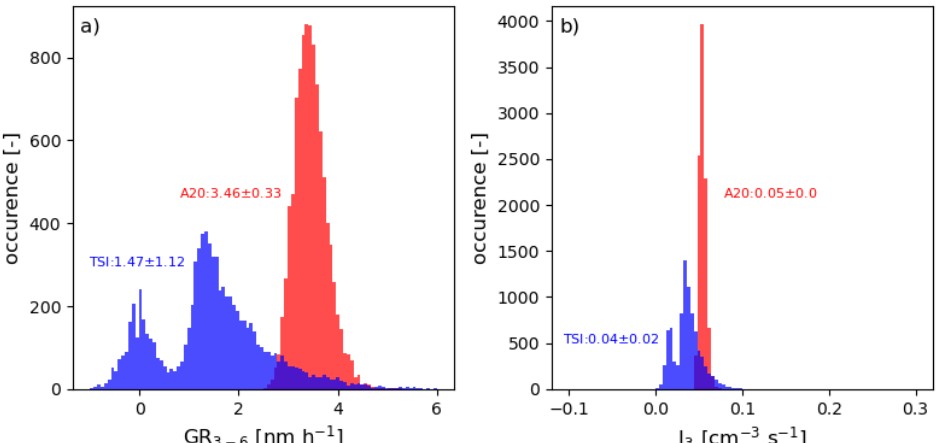

**Figure 7.** Results from the Monte Carlo simulations testing the influence of a pure counting error and an additional measurement error on the size distribution-derived quantities $J_3$ and $GR_{3-6}$ for a very weak NPF event ($J_3 \sim 0.05$ cm$^{-3}$ s$^{-1}$) **.T**he histograms of the Monte Carlo results for the $GR_{3-6}$ (a) and $J_3$ (b) are shown. The red histograms correspond to values derived from the modified Airmodus A20 data assuming only a counting error and the blue histograms correspond to values derived from the TSI 3776 data assuming only a counting error.

## 5.3 Estimating the total error of the TSI 3776 and its effect on the particle formation and growth rates

We now aim to estimate the total error in a CPC measurement based on our dual setup. As described by Eq. (6), we can obtain an upper estimate of the total error in the TSI 3776 measurement by selecting small count ranges in the modified Airmodus A20 and estimating the width of the resulting count distribution in the TSI 3776 at simultaneous measurements. In Fig. 8a we show that upper relative error estimate together with the pure counting error ($\sigma = \sqrt{N}$) for a set of selected count intervals in the modified Airmodus A20 versus the expected value of counts in the TSI 3776 ($E[N_{TSI}] = Q_{TSI}/Q_{A20} \cdot N_{A20}$). We see that the relative uncertainty is significantly larger than what would be expected from a pure counting error, indicating that there are also other important sources of uncertainty in a typical DMPS measurement, resulting from fluctuations in flow rates or electronic noise. If we further assume that the relative uncertainty of such an additional source is the same for any CPC, we can further simplify Eq. (6) by setting $\left(\frac{\Delta N_{CPC}^{meas}}{N_{CPC}}\right)^2 = \left(\frac{\Delta N_{TSI}^{meas}}{N_{TSI}}\right)^2 = \left(\frac{\Delta N_{A20}^{meas}}{N_{A20}}\right)^2$ and even solve for that missing error source, which is shown in Fig. 8b. We obtain a roughly constant value of around 4% across all count ranges, also indicating that these fluctuations are indeed independent from the counting error.

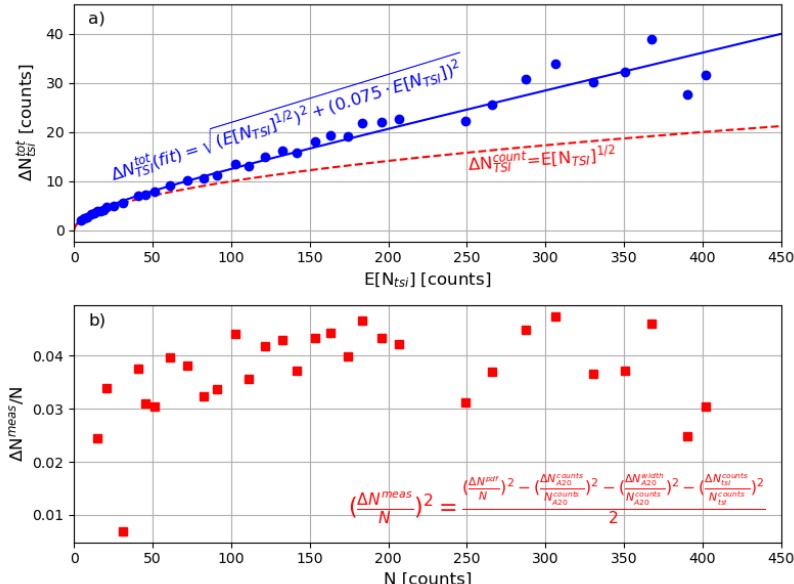

**Figure 8** Total uncertainty estimate for the TSI 3776 by selecting narrow count ranges in the modified Airmodus A20. (a) shows the estimates of the total uncertainty via Eq. (6) for several count ranges in the modified Airmodus A20 as blue circles. The blue line is a fit describing the total uncertainty as the quadratic sum of the counting uncertainty and a measurement uncertainty with its relative magnitude being the free parameter of the fit. The red dashed line shows the pure counting uncertainty as reference. (b) shows the relative measurement uncertainty as red squares when Eq. (6) is solved under the assumption that the relative uncertainty in both CPCs has the same magnitude (see in-panel equation).

To estimate the influence of such additional uncertainties in CPC measurements on the size distribution-derived quantities $GR_{3-6}$, $GR_{6-10}$ and $J_3$, we performed another Monte-Carlo simulation using a fitted expression as in Eq. (6) (counting uncertainty plus an additional measurement uncertainty, where its relative magnitude is the free parameter of the fit) to the total error in Fig. 8a as the input for the variation of the measured counts in the TSI 3776. Fig. 9 shows the resulting histograms for $GR_{3-6}$, $GR_{6-10}$ and $J_3$ comparing it to the pure counting uncertainty Monte Carlo simulation. While the distributions are even further skewed, the relative widths do not dramatically increase further (up to 23% relative uncertainty on $GR_{3-6}$). However, as the relative counting error is so much lower in the modified Airmodus A20, we suspect that this additional source of uncertainty would dominate the formation and growth uncertainties in that case by the following simple reasoning: the relative counting uncertainty scales with $1/\sqrt{N}$ and the measurement uncertainty seems to be independent of the number of counts (Fig. 8b), and hence the ~4% measurement uncertainty start to dominate the total uncertainty above 625 counts as $1/\sqrt{625} = 0.04$, which is roughly the sub-5 nm count rates measured in the modified Airmodus A20 during the NPF event of the 28th March 2017.

Our limited dataset does not allow for the reverse procedure due to a lack of statistics (i.e. selecting narrow count ranges in the TSI 3776 and obtaining the pdf for the simultaneous measurements of the modified Airmodus A20) and hence we do not provide a detailed Monte Carlo analysis on the effects on the growth and formation rate. However, the analysis of the additional event at reduced $J_3$ (Fig. S2 in the Supplement) at least reveals that at even lower counting uncertainties, the influence of the





measurement error on the size distribution-derived quantities $GR_{3-6}$ and $J_3$ becomes negligible as almost no further broadening of the Monte Carlo result distributions is observed upon inclusion of this additional uncertainty.

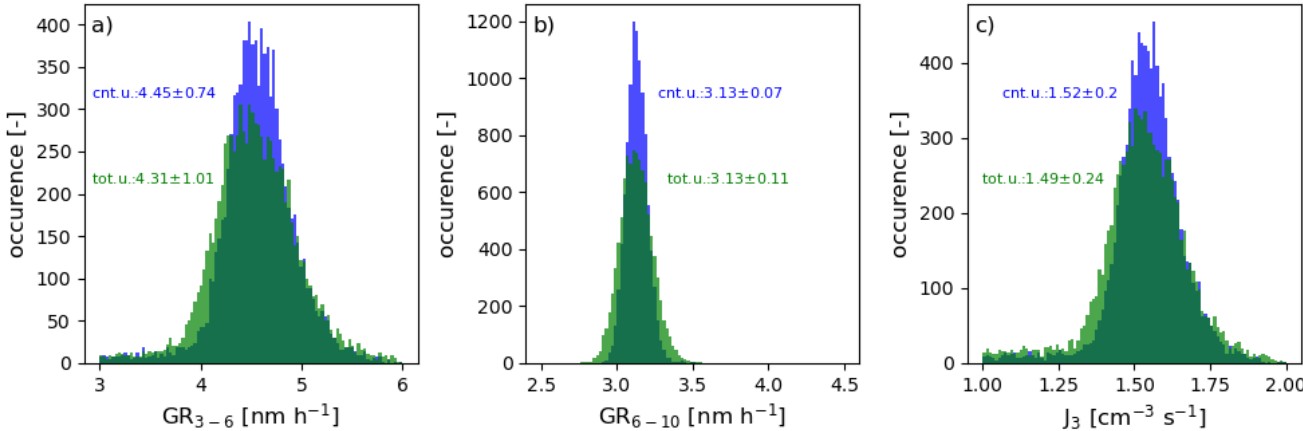

**Figure 9** Results from the Monte Carlo simulations testing the influence of a total measurement error in the TSI 3776 on the size distribution-derived quantities $J_3$, $GR_{3-6}$ and $GR_{6-10}$. (a) shows the Monte Carlo outcomes for $GR_{3-6}$ using only the counting statistics as variation for the input data from the TSI 3776 in blue (same as Fig. 6d) and using the total uncertainty in green as derived via Eq. (6) and the fit from Fig. 8. (b) shows the same for $GR_{6-10}$, and (c) for $J_3$ using the same color convention.

## 6 Conclusions

The strength and importance of NPF with respect to the climate system is often characterized by formation and growth rates, which are commonly derived from the evolution of measured particle number size-distributions obtained from DMPS/SMPS systems. However, the uncertainties in the DMPS measurements and its effect on the size distribution-derived quantities are not well quantified. As the CPC counting process can be considered as a Poisson process, the resulting uncertainty from the counting process can be non-negligible at the low count rates in the sub-10 nm range and might dominate the uncertainty in the derived size-distribution and formation and growth rates.

Here, we deploy a DMPS system with a modified Airmodus A20 CPC providing a factor 50 higher counting statistics compared to the commonly used TSI 3776 ultrafine CPC. We found that the modified Airmodus A20 provides smoother number size-distributions, especially in the case of low concentrations of ultrafine particles and achieves very good correlation with simultaneous absolute number concentration measurements. The difference between the counting statistics of the CPCs is propagated to the values derived from the measured number size distribution, resulting in significantly reduced uncertainties for $GR_{3-6}$ (1% compared to 16%), $GR_{6-10}$ (0.3% compared to 2%) and $J_3$ (1% compared to 13%). This effect is even stronger, when the formation rates and hence number concentrations are low, where a reliable GR estimate might only be possible with a DMPS with sufficient counting statistics. In addition, our dual CPC DMPS setup allowed for a quantification of the total uncertainty related to the CPC measurement in a DMPS system, showing that additional sources of uncertainties with a relative uncertainty of around 4% are present at all count rates. These sources can additionally contribute to the uncertainties of the





size distribution-derived quantities and might be more and more important as the counting uncertainties are reduced as in the case of the modified Airmodus A20.

This study shows the significant improvement in the determination of the formation and growth rate during NPF by the deployment of a DMPS with improved counting statistics. The wide deployment of such instrumentation which is optimized for sub-10 nm measurements could significantly reduce our uncertainties in formation and growth rate determination or even

allow for the application of better analysis tools due to the increased statistics (Pichelstorfer et al., 2018; Ozon et al., 2021) and hence boost our understanding of NPF, e.g. the provide better mass closure in aerosol growth (Stolzenburg et al., 2018). However, this study also shows that other sources of uncertainty are typically present in DMPS measurements, which also need to be understood and potentially be reduced or at least be well-quantified, which requires future work on CPC techniques.

**Acknowledgements**

We thank Lubna Dada for her support in nucleation rate calculations. This work was funded by the Academy of Finland Flagship via the Atmosphere and Climate Competence Center (ACCC), grant number 337549 and the Academy of Finland grants 1325656, 346370 and 79999129. It also received funding from the University of Helsinki three-year grant (75284132) and the University of Helsinki ACTRIS-HY funding. It also received support from the European Union's Horizon 2020 research and innovation program under Marie Skłodowska–Curie grant agreement no. 895875 (NPF-PANDA), from the

European Commission through Research Infrastructures Services Reinforcing Air Quality Monitoring Capacities in European Urban & Industrial AreaS (RI-URBANS), grant no. 101036245. and through ACTRIS-CF (329274) and ACTRIS-Suomi (328616).

**Competing interests**

Joonas Vanhanen is the Chief Technology Officer of Airmodus Ltd., the company producing and selling the A20 CPC. The
remaining authors have no conflicts of interest to declare. This study was independently performed and was not co-funded by Airmodus Ltd.

**Author contributions**

T.L., P.A., J.V., J.K. performed the measurements, D.S., T.L. analysed the data and performed the simulations, D.S., T.L., T.P., J.K. were involved in the scientific discussion and interpretation of the results, D.S., T.L. wrote the manuscript, all co-
authors commented on the manuscript.





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
