# Peer review of "Improved Counting Statistics of an Ultrafine DMPS System"

_Atmospheric Measurement Techniques, 2022_

## Referee Comment (RC1)

Comment on amt-2022-270

General comments:

The present paper focuses on improving the counting statistics of sub-10 nm aerosol particles using a DMPS system with a modified Airmodus A20 CPC. They further found that improving counting statistics significantly reduces the uncertainty with the estimation of new particle formation and growth rates. I found it very important and interesting for aerosol size distribution measurements. The manuscript was well-written and structured. I would suggest publishing it on AMT after a minor revision.

Specific comments:

1) Line 92. Why do you choose 2.5 lpm? Since a higher inlet flow rate indicates higher counting statistics, can you increase even higher?

2) Section 2.4 uncertainty in CPC measurements. As so many uncertainties are calculated (e.g., counting, measurement, total…) and discussed later in the results section, it is sometimes difficult to follow for readers. I suggest making a summary table listing all uncertainties, including the formula, use of purpose and values for exemplified experiments (e.g., 28$^{th}$ March, 5$^{th}$ and 6$^{th}$ May).

3) Line 142-143 and Figure 2. Why do you choose these certain number ranges?

4) Line 170, …fits sigmoidal functions to the rise of the measured signal.. of which parameter (number concentration)?

5) Line 180-182. What does CS mean in equation (8)? Give some details on how to calculate GR$_{3-6}$. It would be helpful to have exemplified fitting plots to derive GR and J3 for non-NPF background readers.

6) Line 186. Why do you choose 28 March as an example?

7) Line 187-188. By altering the measured counts in each size-channel for each measurement time according to their underlying uncertainties. Plot out the time series of the uncertainties or give numbers (e.g., avg. +/- std)

8) Line 222-224. How much does the chemical composition influence the cut-offs? Give a number if available.

9) As shown in figure 4, what is the meaning of the numbers: 0.94-70.0 and 0.89-14.0?

10) Line 236-237. Why the formation rate is more robust even though the used GR$_{3-6}$ is less consistent between the A20 and TSI 3776? It would make sense if the GR term is not the dominant term. But in lines 268-269, the author demonstrated that the dominant term for formation rate is the growth term.

11) Line 241-242, Figure 6 is not well described and explained, add more details if you think it is important otherwise delete it. In Figure 6 (c), formation rate for A20 or TSI 3776? In Fig.6 d-f, why the distributions of A20 are always narrower than TSI?

12) Line 253, please clarify how you derive the statistical uncertainty, refer to the table in comment 2.

13) Figure 7 (a), any explanations on why is the distribution of GR$_{3-6}$ bi-modal?

14) Lines 277-281, please clarify the relative uncertainties, refer to the table in comment 2.

15) Out of curiosity, is it possible to compare A20 with NAIS as NAIS is good with nano-particles down to 0.8nm?

---

## Author Comment (AC1)

**Referee #1:**

The present paper focuses on improving the counting statistics of sub-10 nm aerosol particles using a DMPS system with a modified Airmodus A20 CPC. They further found that improving counting statistics significantly reduces the uncertainty with the estimation of new particle formation and growth rates. I found it very important and interesting for aerosol size distribution measurements. The manuscript was well-written and structured. I would suggest publishing it on AMT after a minor revision.

We thank the Referee for their thoughtful comments which helped to improve the manuscript. Please find our detailed responses below in blue color and changes to the manuscript in red.

Specific comments:

1) Line 92. Why do you choose 2.5 lpm? Since a higher inlet flow rate indicates higher counting statistics, can you increase even higher?

The referee is indeed correct, that higher flow rates than 2.5 lpm would provide even better counting statistics. However, increasing the flow rate results in two challenges. First, the necessary supersaturation for activation of small particles becomes more difficult as the inlet air might not be fully saturated within the current design of the A20. Kangasluoma et al. (2015) indeed showed that the activation efficiency starts to decrease already at these flow rates. Further optimization of the CPC design would therefore be required to achieve even higher flow rates at similar activation properties. Second, higher flow rates would also require higher sheath flow rates in the DMA to maintain the same resolution. While DMAs with much higher sheath flow rates exist, they often provide access only to much more limited size-range as the voltage to diameter relation depends on the absolute value of the sheath flow. With voltages being limited to the order of 10 kV to avoid arcing, this sets natural limits to DMAs with high sheath flow rates, good resolution, and a broad accessible size-range, which in turn limits the available flow rates for a detector downstream of the DMA.

We tried to clarify these two points by adding the following sentence on line 92:

"While higher detector flow rates would result in even better counting statistics, it would require adjustments in the CPC design to achieve similar particle activation due to lower supersaturations and would also result in a lower size resolution for the DMPS system if the sheath flow rate remains constant (higher sheath flow rates would in turn reduce the dynamic size-range of the DMPS)."

2) Section 2.4 uncertainty in CPC measurements. As so many uncertainties are calculated (e.g., counting, measurement, total…) and discussed later in the results section, it is sometimes difficult to follow for readers. I suggest making a summary table listing all uncertainties, including the formula, use of purpose and values for exemplified experiments (e.g., 28th March, 5th and 6th May).

We agree with the referee that an overview of the results for the three measurement days would be beneficial, and we added the requested Table to the manuscript in the results section.

3) Line 142-143 and Figure 2. Why do you choose these certain number ranges?

We agree with the referee that this deserves a more detailed explanation. We choose to use a count interval such that it assures good balance between a narrow enough interval (keeping the relative error of the width below 5 % and enough counts per bin such that the resulting count distribution of the TSI is fitted well.  To explain this we added the following sentence to the manuscript:

"This approach of choosing finite count intervals from the Airmodus A20 data instead of just using a single count value is due to the otherwise limited statistics which would not allow for solid fits of the corresponding count distributions of the TSI 3776."

and later we added:

"(…) and the finite width of selected counts in the interval range (with the relative error due to this kept below 5% by our interval selection $N_2=1.05 \cdot N_1$)"

Line 170, …fits sigmoidal functions to the rise of the measured signal.. of which parameter (number concentration)?

Yes, the sigmoidal fits can be applied even to the raw number concentration as they are independent of the absolute magnitude (see Lehtipalo et al., 2014). We clarified this at this point and removed it from the sentence on line 189-191:

"(…) fits sigmoidal functions to the rise of the measured raw number concentration (the approach is independent of the absolute magnitude of the signal and hence the inversion procedure, see Lehtipalo et al., 2014) in each size channel separately."

"The generated input data (counts) were used to directly calculate $GR_{3-6}$ and $GR_{6-10}$ as the appearance time method can be performed on the raw signal and is independent from any inversion procedure (Lehtipalo et al., 2014)."

5) Line 180-182. What does CS mean in equation (8)? Give some details on how to calculate GR3-6. It would be helpful to have exemplified fitting plots to derive GR and J3 for non-NPF background readers.

We thank the reviewer for pointing out that lack of definition here. CS is condensation sink and the abbreviation is now introduced in the text such that it can be understood in the equation. For the detailed explanations of GR and J, we are convinced that Section 2.5 provides the necessary details, and the exemplified plots are already given in Fig. 6a-c.

6) Line 186. Why do you choose 28 March as an example?

We agree that this deserves a short explanation. We chose the 28th March because it is classified as a strong NPF event with good data coverage and an average growth rate such that the nucleation mode persists for a significant time. We therefore added the following to the manuscript:

"The 28th March is chosen as the example day as it is a typical class-1 NPF event day with a strong nucleation rate, but not much higher than average GR, such that the nucleation mode persists over long enough time in the sub-10 nm range to investigate the effect of improved counting statistics in full detail."

7) Line 187-188. By altering the measured counts in each size-channel for each measurement time according to their underlying uncertainties. Plot out the time series of the uncertainties or give numbers (e.g., avg. +/- std)

We agree with the reviewer that it would be nice to see the variation of the uncertainty in each channel over the course of a day and hence added a new Figure to the Supplement, which shows the relative uncertainty as a surface plot for each diameter and its evolution during the 28th of March for all three simulation cases (Airmodus A20 counting uncertainty only, TSI 3776 counting uncertainty only and TSI 3776 total uncertainty). We added the following sentence at the end of Section 2.5 to the manuscript:

"The relative uncertainties for each size-distribution evolution measurement (in time and size) used as input for all three Monte Carlo simulations are shown in Fig. S2 in the Supplement."

8) Line 222-224. How much does the chemical composition influence the cut-offs? Give a number if available.

Wlasits et al. (2020) showed that the variation of the $d_{50}$ cutoff diameter between different seed materials for the unmodified Airmodus A20 is up to 4 nm, and up to 1.5 nm for the TSI 3776. We thus modified the text as follows:

"(…), which can be more than 3 nm difference for the $d_{50}$ cutoff diameter between different seed materials for the unmodified Airmodus A20 (and only 1.2 nm maximum variation for the TSI 3776) (Wlasits et al., 2020)."

9) As shown in figure 4, what is the meaning of the numbers: 0.94-70.0 and 0.89-14.0?

These numbers represent the slope and y-axis offset of the fit, but the reviewer is correct that this should be written differently to be understandable. We further added an explanatory sentence to the Figure caption.

10) Line 236-237. Why the formation rate is more robust even though the used GR3-6 is less consistent between the A20 and TSI 3776? It would make sense if the GR term is not the dominant term. But in lines 268-269, the author demonstrated that the dominant term for formation rate is the growth term.

It is indeed an interesting remark and the reason for this is not entirely clear. The J calculation seems to have some buffering against the GR fluctuations which either comes through the less dominant terms (CoagS, dN/dt) or the number concentration $N_{3-6}$ which also needs to be included in the growth term. However, our findings are somewhat consistent in that sense, as also in the MC simulations a 16% relative uncertainty on the $GR_{3-6}$ value only translated into a 13% relative uncertainty on the $J_3$. N, GR and CS are all highly linked quantities. Fig. 5a shows that $GR_{3-6}$ derived by the A20 is often faster when above 3 nm h$^{-1}$ (where the GR term in the formation rate calculation is more dominant), but at the same time Fig. 4 shows that number concentrations measured by the A20 are slightly lower. This might be one reason why $GR*N_{3-6}$ and hence $J_3$ (dominated by $GR*N_{3-6}$) fluctuates less between the two instruments. We added that little speculation to the text.

"However, as shown in Fig. 4, the modified Airmodus A20 measured slightly lower concentrations compared to the TSI 3776, while $GR_{3-6}$ was measured higher by the Airmodus A20 for values above 3 nm h$^{-1}$. Therefore, in these cases with a high growth term ($\frac{GR}{\Delta dp} N_{dp}$) possibly dominating the formation rate calculations due a fast growth rate (>3 nm h$^{-1}$), the lower $N_{3-6}$ might compensate for the higher $GR_{3-6}$ reducing the fluctuations between the two instruments. In addition, the other terms ($\frac{dN_{dp}}{dt}$ and $CoagS_{dp}N_{dp}$) in Eq. (7) might also buffer the higher GR due to lower $N_{3-6}$ values in that case. "

11) Line 241-242, Figure 6 is not well described and explained, add more details if you think it is important otherwise delete it. In Figure 6 (c), formation rate for A20 or TSI 3776? In Fig.6 df, why the distributions of A20 are always narrower than TSI?

We understand the reviewer's confusion here and apologize that our references to that Figure were wrong. Fig. 6 is indeed already described, but it was referenced as Figure 7. We adjusted that. We also added the information that we used the 3776 for Fig. 6c. In Fig. 6 d and f, the distributions of the A20 are narrower than those of the TSI due to the better counting statistics. This was already explained in the text.

12) Line 253, please clarify how you derive the statistical uncertainty, refer to the table in comment 2.

We added the information to the Figure caption.

13) Figure 7 (a), any explanations on why is the distribution of GR3-6 bi-modal?

As already mentioned in line 280 we attribute the bimodal distribution to problems in the automated fitting. The bimodal distribution could perhaps be reduced requiring strict goodness of fit conditions for an MC result to be accepted, but it is beyond the scope of this work to find an extremely robust automated GR fitting method.

14) Lines 277-281, please clarify the relative uncertainties, refer to the table in comment 2.

As requested by the reviewer, we added the new Table 1 to the manuscript.

15) Out of curiosity, is it possible to compare A20 with NAIS as NAIS is good with nano-particles down to 0.8nm?

Kangasluoma et al. (2020) showed that NAIS in particle mode (which only extends to down to 2.5 nm) still shows significant discrepancies in total concentration measurements compared to other instruments in the sub-10 nm range. It is therefore not ideally suited to calculate formation rates. Growth rates derived from NAIS have been compared extensively with other approaches (e.g. Gonzalez-Carracedo et al., 2022, ACP) showing similar scatter as the two DMPS approaches. As the NAIS measures current and not particle counts, its underlying uncertainty treatment needs to be different and is out of scope of the manuscript.

---

## Author Comment (AC2)

**Referee #2:**

I congratulate the authors for the nice manuscript, and I would like to start by offering my sincere apologies for the delay in providing this review. The authors compared a standard ultrafine CPC with a modified Airmodus A20 CPC to investigate the effect of poor counting statistics on the calculation of growth and formation rates. Uncertainties on these quantities are often neglected and this work offers new and interesting results. The manuscript is well-written and fits the scope of the journal. The analysis is sound, and the results are presented clearly and concisely. However, some minor comments need to be addressed, and the manuscript's clarity can also be improved in a few places.

We thank the reviewer for their insightful suggestions (original comment in black) and please find our responses below (in blue) and changes to the revised manuscript in red.

Minor comments:

Lines 42-44: This sentence about technological development can probably be used for every measurement system. Consider removing or rephrasing it.

We agree with the reviewer and adjusted the sentence to:

"However, there are still significant discrepancies between different particle size-distribution data sets, especially for the sub-10 nm size-range (Kangasluoma et al., 2020)"

Line 44: "a large fraction" can you provide more quantitative information? Losses will clearly depend on the instrumental setup but please mention what the typical range is.

We thank the reviewer for asking to be more quantitative here and put (typically >95%) in the text to give the order of magnitude, based on e.g. Stolzenburg et al. (2017), AMT.

Line 48: The "PI" parameter is not used in the rest of the manuscript, and I do not see the reason for mentioning it here in the introduction. The only relevant information is that a system with lower losses, higher flow rate and sampling time will have better counting statistics. You already explained this in lines 47-48, so you can remove this part on the PI parameter.

We agree with the referee and removed that paragraph.

Line 51: "the PI parameter … describes the instrument sensitivity towards low number concentrations" instrument sensitivity represents the smallest absolute amount of change that can be detected. So, sensitivity should be the same at low and high number concentrations. I think speaking of the signal-to-noise ratio is more meaningful in this context.

Paragraph about the PI-parameter is removed, so no need to change the wording here.

Line 54: Is there any drawback in using a CPC with a higher aerosol flow? I guess there must be an optimum range; otherwise, why not use a 10lpm flow rate or higher? I could think of coincidence (probably not a big issue when the CPC is used behind a DMA), a higher DMA sheath flow is required to keep the same resolution, and probably some other technical issues with the CPC construction itself. Adding a short sentence on the cons/problems of having a larger flow would be useful.

Please see our response to referee #1 concerning this point.

Section 2.2:

- did you also intercompare the counting efficiency of the two CPCs? From Figure S1, it seems there is a ~5% difference; if not corrected, can this affect your results?

The counting efficiency curves as shown in Fig. S1 are included in the inversion and hence the 5% difference is always accounted for when inverted data are used. For growth rate calculations inverted data is not needed as only the relative rise of the signal is considered, which we clarified again in line with the comment of referee #1.

- What about the response time of the two CPCs? This is a relevant parameter for your analysis, so it would be important to report it here (CPC3776 response time is well characterized, but I don't know if it is the same for the modified A20).

  We agree with the reviewer that the response time could be an issue (~0.2 s for the TSI2778 and ~1s for the A20). However, in our entire analysis we compare the counts during a measuring period at a fixed DMPS voltage (i.e. one size), which is always longer than 3.5 seconds for each DMPS step. To get to the total number of counts in that period we took the average concentration and multiplied it by the flow rate and time interval. Over these intervals the different response times should not play any role, especially as the DMPS has a short stabilizing time interval (1 second) before every new measurement, such that we can assume that during the measurement at one size, the number of counts at both CPCs should not be affected by differences in response time. We agree with the referee that this should be clarified and added the following explanation to the manuscript.

  In Section 2.2 we added:

  "Apart from their differences in activation efficiency and effective detector flow rate, the two CPCs have different response times to a change in aerosol concentration, which are ~0.1 s for the TSI 3776 and ~1 s for the (unmodified) Airmodus A20 (Enroth et al., 2018). However, as we will see below that small difference does not affect our approach in comparing the counting statistics of the two CPCs."

  In the newly drafted Section 2.3 we added:

  "In our DMPS, the voltage is stepped from 3 to 1000 V in 17 steps (corresponding to selected mobility diameters of 2.07 to 40 nm assuming singly charged particles), with a settling time of 1 second at the beginning of each voltage step (which should remove any bias from different response times of CPCs, if they are ≤1s). The measured particle number concentration C (in cm$^{-3}$) for size is determined by the number of particles N counted in the time interval $\tau$ where the voltage is kept constant (which varies between 3.5 seconds for the largest size and 64 seconds for the smallest size) a specific measurement volume and the number concentration C can be calculated by using the volumetric flow rate through the optics $Q_{opt}$: (…)"

Line 110: here, you define 'time t' but in equation (1) it is defined as tau; please use the same definition.

Thanks for noting. We now consistently use $\tau$ throughout the manuscript.

Lines 118-125: this part is confusing because you provide the Poissonian distribution before saying what a Poissonian process is. Additionally, you are mixing the general description of a Poissonian process with its applicability to an optical counting system. For example, coincidence applies to certain types of measurement systems but not to a general Poissonian process. My suggestion is first to define a general Poissonian process, then describe the Poissonian distribution and conclude with the applicability of Poissonian statistics to a counting system like a CPC.

We agree with the reviewer that this part is not well-written and reformulated it:

"A random variable N has a Poisson distribution with the parameter $\mu\tau > 0$, where $\tau$ is the measurement time, and $\mu$ is the intensity (rate) of the process, if the random variable can obtain discrete values (0,1,2,3,…) within the time interval $\tau$. If the process is characterized by the following

properties: 1) For $\tau = 0$ we have $N(0) = 0$, 2) in separate time intervals, the numbers of detected events are independent of each other and 3) the number of events in any interval of length $\tau$ obey the Poisson distribution:

$$P(N(\tau) = N) = \frac{e^{-\mu\tau}(\mu\tau)^N}{N!} \qquad (1)$$

A Poisson distribution can be shown to have the following properties: the expected value $E[N]$ of the distribution can be calculated as $E[N] = \mu\tau$, and the standard deviation ($\sigma$) can be calculated as $\sigma = \sqrt{VAR[N]} = \sqrt{\mu\tau} = \sqrt{E[N]}$.

In a CPC, the particles are counted in the optical unit of the CPC, where a nozzle directs the particle stream to cross a laser beam perpendicularly. Light is scattered from the laser beam as the particles cross it, and the scattered light is collected by a photodiode. In typical optics with ~1 lpm aerosol flow, the probability of coincidence in the counting process is negligible with moderate number concentrations ($< 30\ 000$ cm$^{-3}$), which are typically measured downstream of a DMPS system. In our DMPS, the voltage is stepped from 3 to 1000 V in 17 steps (corresponding to selected mobility diameters of 2.07 to 40 nm assuming singly charged particles), with a settling time of 1 second at the beginning of each voltage step (which should remove any bias from different response times of CPCs, if they are $\leq$ 1s). The measured particle number concentration $C$ (in cm$^{-3}$) for size is determined by the number of particles $N$ counted in the time interval $\tau$ where the voltage is kept constant (which varies between 3.5 seconds for the largest size and 64 seconds for the smallest size) by using the volumetric flow rate through the optics $Q_{opt}$:

$$C = \frac{N}{Q_{opt}\cdot\tau} \qquad (2)$$

If we assume that the number concentration remains constant during the voltage scan of the DMPS (which is anyways also a requirement for any inversion procedure which considers multiply charged aerosols), the counting process in the DMPS can be considered a Poisson process.

In our setup, we can neglect the total penetration of the system since the compared CPCs measure in parallel in the same DMPS system and the total penetration is the same for both. This allows us to compare the raw data from the CPCs without an inversion and the uncertainties related to it (Stolzenburg et al., 2022). As our DMPS outputs the average concentration during each voltage step, we need to rearrange Eq. (2) for the counted particles $N$. This also shows that we can predict that a factor 50 increase of $Q_{opt}$ (effective undiluted optics flow of 0.05 lpm in the TSI 3776 versus 2.5 lpm in the modified Airmodus A20) should lead to a factor 50 increase of N:

$$N = C \cdot \tau \cdot Q_{opt} \qquad (3)"$$

Lines 126-129: when reading this part, I was confused about the applicability of Poissonian statistics to your problem because N during an NPF event is a function of time and is not Poissonian. It is easy to see this if you think that for Poisson P(t1) = P(t2) for any t1 and t2 but for NPF P(t1)<P(t2) if t2>t1 (the particle number increases with time during NPF). After reading the manuscript, I understood that this is probably not a concern because you are working with narrow concentration intervals where the time dynamics likely do not play a role. However, I think it is necessary to comment on this and on the general applicability of Poissonian statistics to describe NPF events.

We thank the reviewer for thinking about the applicability of the Poisson process in that respect. We think that it is the assumption of a constant size-distribution and hence CPC inlet concentration during one voltage step of the DMPS (hence size) which justifies the applicability of the Poisson distribution. Within that measurement interval, two separate time intervals would be independent of each other.

It does not matter if during a day the measured concentration changes, as each measurement by itself is a Poisson process. We clarified this in the newly drafted Section 2.3:

"If we assume that the number concentration remains constant during the voltage scan of the DMPS (which is anyways also a requirement for any inversion procedure which considers multiply charged aerosols), the counting process in the DMPS can be considered a Poisson process."

Line 132: Any observed system is characterized by random fluctuations leading to some sort of inherent variability, but I would not classify this as 'random uncertainty'. I attribute random uncertainty to fluctuations in the measuring system (e.g., small changes in the flow rate, laser current,…).

Lines 142-145, a few comments regarding this approach:

- As mentioned before, what is the effect of the instrument response time? I would expect that if the response time is substantially different, then, with this approach, you would amplify the error. However, this is not a real error because of the different instrument transfer functions. Ideally, you should account for it before performing this analysis (especially considering that you are working with narrow time intervals).
  We thank the reviewer for his careful comment and please see our response above. The response time is no issue here, as we compare counts within time intervals of constant voltage which are significantly larger than the CPC response times of both instruments.
- Instead of considering a narrow interval, why didn't you select a single count value (e.g. pick 300 counts per unit time in the A20 and compare it with the corresponding distribution in the 3776 CPC)? This would remove the uncertainty related to the finite interval selection in the A20.
  We needed to choose a narrow interval as otherwise there would not be sufficient events at a single count value in order to obtain reasonable fits. As also requested by referee #1, we clarified this in the text, please see our response there.
- Mention explicitly that a gaussian distribution is a good approximation of a Poissonian when mu*tau is sufficiently large (>~10), which is why you are using a gaussian PDF to fit the data.
  Agreed. We added that to the text.
  "(which is a good approximation to a Poisson distribution when $E[N] > 10$)"

Line 188: Did you use the square root of N as underlying uncertainty? If so, please mention it explicitly.

We added that.

"(assuming a $\sqrt{N}$ uncertainty)"

Line 192: Replace the second "using" with a different verb and the correct form (e.g. considers).

Thanks. Corrected to "considers".

Lines 219-222: You could make the same scatter plot for periods with no NPF to exclude the sizing effect. It should be an easy check with your dataset and would probably resolve this open question (Fig. S1 already shows a difference in the counting efficiency).

This is an interesting idea. We checked it and could not find a significant improvement for the agreement when the NPF days are removed from the dataset. Therefore, we removed our statement about the cutoff uncertainty from the text.

Figure 5: to what extent can the Poissonian statistics explain the observed discrepancies in GR and J? I guess that for a quantitative answer, you would have to run the MC simulation for all events, which is not what I am asking for, but a comment on this aspect would be useful for the paper.

We thank the reviewer for that interesting question. We now indicate the counting error of the GR3-6 and J3 measurements in Fig. 5 for all three example days. Within these errorbars the obtained GR and J values all fall indeed onto the 1:1 line, which gives some indication that the observed scatter can be explained by the uncertainty from the counting error. We added the following text:

"Altogether, the counting uncertainties derived for all three days analyzed by the Monte Carlo approach can explain the observed scatter between the values derived by the two instruments (see errorbars on the three selected events in Fig. 5), which implies that the counting uncertainty is a major issue when GR and J values are compared between different instruments."

Figure 6 and Figure 7: is the GR and J distribution centered around the "real" value? It would be useful to report the value measured for the real event (or mention that is the same as the distribution mean if this is the case).

We agree with the reviewer that this information should be added to the manuscript. For the strong NPF day, the distributions are indeed well centered around the original result obtained from the actual measurement data, which we added to the text:

"(…) demonstrating the observed variations shown in Fig. 5 and with the mean of the distributions roughly centered around the original result."

For the weaker NPF days this changes, especially for the 5$^{th}$ of May, where the distribution is significantly offset from the result obtained from the original data. We thus added:

"At very low J3 (5$^{th}$ May 2017, Fig. 7), the Monte Carlo distributions for the TSI 3776 data get skewed (with the mean of the distribution also deviating significantly from the original result) and (…)"

Line 263: is the statistical uncertainty defined as one standard deviation? Please report which type of statistical uncertainty was used.

Agreed. We added "(defined as 1σ standard deviation of the Monte Carlo derived distribution divided by the initial GR$_{3-6}$ result obtained from the actual measurement data)".

Line 281: "with" instead of "which".

Changed. Thanks.

Line 295: "the" instead of "that"

Changed. Thanks.

Section 5.3: This part is interesting because it shows that counting statistics is the main source of uncertainty for GR and J determination. You show that other CPC measurement errors can be neglected even with your upper-limit approach (you are essentially attributing all A20 measurement errors to the TSI CPC). However, this message is not very explicit, my suggestion is to restructure this section to clarify this point. I think this is an important conclusion and should be highlighted better.

We agree with the reviewer that this could be highlighted better and added the following text:

"For the events at reduced J$_3$ (Fig. S3 in the Supplement and Table 1) the influence of the measurement error on the size distribution-derived quantities GR$_{3-6}$ and J$_3$ becomes almost negligible compared to the even higher counting uncertainties as almost no further broadening of the result distributions are

observed. Altogether, this clearly demonstrates that the counting uncertainty is the dominant source of error for nucleation and growth rate determination when a TSI 3776 ultrafine CPC is used."

Line 338: I would remove 'sub-10 nm range', your findings regarding Poissonian statistics apply to any size range, and absolute counts are often lower for larger particle sizes.

Agreed and removed.

Lines 349-351: I would rephrase this because, for most practical applications (the majority of NPF studies are performed with UCPC having a small flow rate), this additional source of error seems negligible, as you have shown in the previous section. So, I would say that the additional measurement error becomes important only when using a system with high counting statistics, as in the case of the A20 CPC.

We agree with the reviewer and adjusted these sentences to:

"However, we showed that the counting uncertainty is main source of error for the size distribution-derived quantities J and GR for the widely used TSI 3776. The additional sources of uncertainty might only become important in the derivation of the nucleation and growth rates when the counting uncertainties are reduced as in the case of the modified Airmodus A20."